# Trend and risk factors of fatal pregnancy termination: A long-term nationwide population-based cross-section survey in Bangladesh

**Shathi Das**[1‡]*, **Jui Das**[2‡], **Kamanasis Mazumder**[3], **Palash Roy**[4], **Rehana Begum**[5], **Sumon Kumar Das**[6]*

1 St. Gregory's High School & College, Dhaka, Bangladesh, 2 Mater Research Institute, The University of Queensland, Brisbane, Australia, 3 Ministry of Health and Family Welfare, Dhaka, Bangladesh, 4 Kumudini Pharmaceutical Limited, Dhaka, Bangladesh, 5 Department of Food and Nutrition, College of Home Economics, Dhaka, Bangladesh, 6 Menzies School of Health Research, Charles Darwin University, Casuarina, Australia

‡ These authors share first authorship on this work.
* dasshathi1@gmail.com (SD); dassumonkumar1@gmail.com (SKD)

**Data Availability Statement:** All the survey data (secondary data) are publicly available through the Monitoring and Evaluation to Assess and Use Results Demographic and Health Surveys

## Abstract

### Background

Pregnant women often experience the fatal outcome of their pregnancy both in developed and impoverished countries. Due to strong health systems and services, factual and historical data are available from developed countries. However, the prevalence trend and risk factors of a fatal termination of pregnancy in developing countries like Bangladesh are still lacking.

### Objective

The objective of the current study was to determine the 20 years trend of prevalence and risk factors of fatal pregnancy termination from 1997 to 2018 in Bangladesh.

### Method

This study utilised the publicly available seven consecutive cross-data on Bangladesh Demographic and Health Surveys data since 1997 following identical methods among women of reproductive age. Respondent was asked if they had had a fatal pregnancy termination ever. A Generalised Linear model with a log-Poisson link was used to estimate the relative risk of different predictors for four survey time points (1998, 2004, 2011, 2018).

### Results

The proportions of fatal pregnancy termination in urban and rural areas were 24% vs. 19% and 24% vs. 22% in 1997 and 2018, respectively. In multivariable analysis, maternal age 30 years and above and obesity were strongly associated in all survey time points. The richest wealth index had a weak association in 1997 but was strongly associated in 2011 and 2018. A significant modest association with secondary complete education level was only observed in 2018.

(MEASURE DHS). The link is: https://dhsprogram.com/Data/.

**Funding:** The authors received no specific funding for this work.

**Competing interests:** The authors have declared that no competing interests exist.

**Abbreviations:** BDHS, Bangladesh Demographic Health Survey.

## Conclusion

The overall proportions of fatal pregnancy termination in Bangladesh remain nearly static; however, its risk factors differed across different survey time points.

## Introduction

Natural/fatal pregnancy termination is a common event during childbearing age [1, 2]. It is due to genetic or hormonal abnormalities and an abnormal maternal immune response, and several asymptomatic infections are responsible for any fatal event of conception [3, 4]. Poor hygiene practices, maternal nutrition, environmental change are other factors that might be confounding or mediating the unfavourable uterine atmosphere resulting in pregnancy termination [1, 2, 5, 6].

Recent research has identified the aetiology and risk factors of spontaneous abortion or miscarriage and stillbirth [2, 7]. Nevertheless, fatal pregnancy termination remains a sustained concern in terms of high prevalence in developed and impoverished countries [1]. Globally, 39 abortions occur among every 1,000 women of childbearing age [1]. On the other hand, data published in 2015 indicated an estimated 2.6 million third trimester stillbirths occurred globally [8]. The prevalence of pregnancy termination may differ across ethnicity and social classes [1] and be aggravated under specific predisposing co-morbidities such as obesity, asthma, diabetes, and hypertension [9–12]. However, the preventive strategies for the natural termination of pregnancies are still not correctly established [1, 2].

In Bangladesh, fatal pregnancy termination is also a significant concern [13–16]. According to the nationally representative cross-sectional Demographic and Health Surveys, there was no notable change in the prevalence of fatal pregnancy termination to date (1994 to 2018) [15, 16]. However, none of the surveys reported the potential risk factors for termination of pregnancies, particularly any change in determinants of termination of pregnancies as a whole. Thus, the current study aimed to determine the changing trend of the prevalence of fatal pregnancy termination and their risk factors over time by analysing Bangladesh's Demographic and Health Surveys data.

## Method

### Demographic and health surveys

The Bangladesh Demographic and Health Surveys are two-stage stratified sampling design cross surveys conducted since 1993 following almost identical questionnaires and sampling schemes [15, 16]. Sample stratification was carried out first as census enumeration areas (primary sampling units). Then, randomly selected households from these primary sampling units were stratified by urban and rural areas **within** each country's division. The detailed sampling and survey methods were described elsewhere [16]. All the survey data (secondary data) are publicly available, and seven consecutive surveys since 1997 were included in the current study.

### Ethical approval

The Bangladesh Demographic and Health Surveys data are publicly available through the Monitoring and Evaluation to Assess and Use Results Demographic and Health Surveys (*MEASURE DHS*). Data for the current study was downloaded after proper registration in the *MEASURE DHS*. Thus, no ethical approval is required for the current secondary data analysis.

### Primary outcome and exposure of interest

In each survey, information on ever had a pregnancy that miscarried, ended using menstrual regulation, was aborted or ended in a stillbirth was collected and considered the primary outcome (yes = 1 and no = 0).

Several socio-demographic factors were identified and considered as risk factors. Such as: women age grouped as less than 19 years, 19 to 29 years (reference group), 30–39 years and 40 years and above; (ii) a geographical area as urban and rural (reference group); (iii) household pre-estimated wealth-index which classified as richest, richer, middle, poorer, poorest (reference group); (iv) religion grouped as Islam (reference group) and non-Islam due to 90% of the study participants were Muslim; (v) maternal level of education categorised as no education, up to the primary, up to secondary, and higher (reference group); (vi) household floor materials as earth and non-earth (reference group).

Two water and sanitation variables were included. For example, (i) household toilet facility which was broadly grouped as Ventilated Improved Pit latrine (VIP) (reference group), flush toilet, pit latrine with/out a slab and no facility/bush/hanging; (ii) drinking water source tube well or borehole (reference group) vs. other.

In each survey, maternal weight and height were measured. Thus, body mass index was equated by weight in kg. divided by height in meter square. Hence, body mass index stratified following standard classification as under-nutrition ($<18.5$), normal (18.5 to $<25$; reference group), overweight (25 to less than 30) and obese (30 and above) [17].

### Statistical analysis

Complete case analysis was performed, excluding the missing data (non-responsive or refuse or unable to measure), specifically weight and height and other variables.

Both descriptive and multivariable analyses were used to determine the change in predictors of fatal pregnancy termination using the individual survey weight of each survey time point.

We performed analyses pin four survey time points, 1997, 2004, 2011 and 2018, respectively, to estimate the change of predictors over time. In addition, the ratios of fatal pregnancy termination for all explanatory variables were estimated between the first and last survey time point (weighted prevalence in 2018 divided by weighted prevalence in 1997). Finally, Generalised Linear models using a log-Poisson link were used to estimate the association (relative risk and their 95% confidence interval) between different predictors and the fatal pregnancy termination for each of the survey time points.

All statistical analyses conducted using STATA version 15 (Stata Corp, College Station, TX, USA).

## Results

A total of 79739 records were analysed [1997 (N = 4080), 2000 (N = 4829), 2004 (N = 9435), 2007 (N = 9801), 2011 (N = 15838), 2014 (N = 17673), 2018 (N = 18083)]. Although there was a significant difference in the overall weighted proportion of fatal pregnancy termination between urban (24%) and rural (22%) areas, the difference was not that big. In 1997 the proportion of fatal pregnancy termination in urban and rural areas were 24% vs. 19%; while in 2018, they were 24% vs. 22%, respectively (Fig 1).

Table 1 represents a detailed distribution of different factors among women with an ever history of fatal pregnancy termination at four survey time points. There was a—change in maternal age of 40 years and above with fatal pregnancy termination between 1997 to 2018 (ratio between 2018 and 1997 was 4.83). Changes were also observed for the richest wealth-

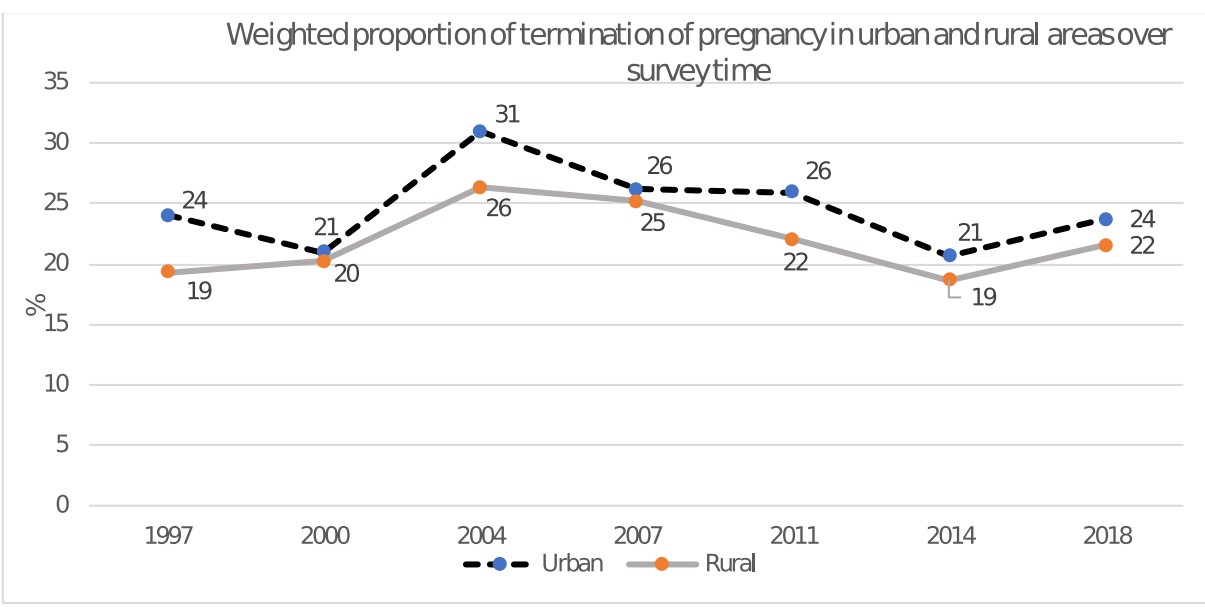

**Fig 1. Weighted proportion of fatal pregnancy termination in urban and rural areas over survey time.**

index groups (1.17), women with no education (2.68), overweight (11.37) and obese (6.95). Similar changes were also observed for women without a history of pregnancy termination between the two-time points.

In multivariable analysis (Table 2), in 1997, the relative risk of maternal age 30 to 39 years (RR: 1.52) and 40 years and above (RR:1.63) were associated with an ever fatal termination of pregnancy, which persistently remains significant in the other three survey time points. The relative risk slight decrease since 1997. Conversely, in 1997 teenage mothers (<19 years) were significantly 68% (RR: 0.32) less likely to have fatal termination of pregnancy which gradually decreased to 36% in 2011 and only 4% in 2018. Maternal obesity was significantly associated with ever fatal pregnancy termination at all survey time points except 2004, and the risk decreased from 2.04 in 1997 to 1.27 in 2018. In contrast, the pregnant women of the richest wealth index had an insignificant, weak association in 1997 (RR:0.92), but relative risk for ever fatal pregnancy termination was significantly associated in 2011 (RR:1.30) and 2018 (RR:1.20). Mother completed education up to secondary level was significantly associated with fatal pregnancy termination observed for 2011 and 2018. Complete primary education level was associated considerably in 2011 only, though the risk was relatively weak (RR:1.12). For other factors, the associations with ever fatal pregnancy termination were mixed and detailed reported in Table 2. For example, significantly 16% (RR:0.84) and 29% (RR: 0.71) less risk and 14% (RR:1.14) excess risk were observed for non-Muslim women in 2011, women of households with the earth as floor material in 1997 and non-tube-well source of dirking water in 2004 only, respectively for fatal termination of pregnancy.

## Discussion

Recent evidence showed that the rate of global unintended pregnancy has reduced. Still, the number of unintended pregnancies ending in abortion has increased over the last three decades [1]. However, in the current study, during the previous two decades, the proportion of fatal pregnancy termination in Bangladesh remained same with some minor variations. Thus, it is difficult to explain the reason for static proportion of fatal pregnancy termination over

**Table 1. Proportion (weighted) [1] of different factors by fatal pregnancy termination in four survey time points.**

| | 1997 | | 2004 | | 2011 | | 2018 | | Ratio between 2018 and 1997 | |
|---|---|---|---|---|---|---|---|---|---|---|
| | No | Yes | No | Yes | No | Yes | No | Yes | No | Yes |
| **Age** | | | | | | | | | | |
| 19 to 29 years | 62.41 | 59.37 | 43.96 | 35.74 | 43.95 | 35.69 | 39.99 | 29.69 | 0.64 | 0.50 |
| <19 years | 15.08 | 3.92 | 7.52 | 3.90 | 4.80 | 2.64 | 3.92 | 2.64 | 0.26 | 0.67 |
| 30 to 39 years | 18.99 | 30.65 | 28.74 | 35.74 | 28.26 | 33.97 | 32.80 | 38.36 | 1.73 | 1.25 |
| 40 years and above | 3.51 | 6.06 | 19.79 | 24.62 | 22.99 | 27.70 | 23.28 | 29.30 | 6.63 | 4.83 |
| **Area** | | | | | | | | | | |
| Rural | 90.05 | 87.26 | 78.26 | 74.17 | 75.52 | 71.36 | 72.70 | 70.14 | 0.81 | 0.80 |
| Urban | 9.95 | 12.74 | 21.74 | 25.83 | 24.48 | 28.64 | 27.30 | 29.86 | 2.74 | 2.34 |
| **Wealth-index** | | | | | | | | | | |
| Poorest | 19.75 | 20.98 | 21.32 | 18.42 | 19.60 | 16.96 | 19.73 | 17.07 | 1.00 | 0.81 |
| Poorer | 22.06 | 21.63 | 20.11 | 20.33 | 20.19 | 18.12 | 20.07 | 19.93 | 0.91 | 0.92 |
| Middle | 17.47 | 18.65 | 20.31 | 17.54 | 20.17 | 19.97 | 20.51 | 20.09 | 1.17 | 1.08 |
| Richer | 20.30 | 19.42 | 19.47 | 20.76 | 20.29 | 20.48 | 20.62 | 20.26 | 1.02 | 1.04 |
| Richest | 20.43 | 19.32 | 18.79 | 22.96 | 19.74 | 24.47 | 19.07 | 22.64 | 0.93 | 1.17 |
| **Religion** | | | | | | | | | | |
| Muslim | 91.15 | 92.11 | 89.66 | 90.82 | 89.63 | 91.31 | 90.14 | 91.28 | 0.99 | 0.99 |
| Non- Muslim | 8.85 | 7.89 | 10.34 | 9.18 | 10.37 | 8.69 | 9.86 | 8.72 | 1.11 | 1.11 |
| **Education** | | | | | | | | | | |
| Higher | 53.82 | 54.46 | 43.27 | 44.24 | 29.50 | 28.87 | 17.62 | 18.03 | 0.33 | 0.33 |
| Upto secondary | 28.20 | 24.30 | 29.74 | 29.28 | 30.73 | 31.47 | 32.33 | 34.45 | 1.15 | 1.42 |
| Complete primary | 15.41 | 17.35 | 22.62 | 22.18 | 33.56 | 33.36 | 39.41 | 37.14 | 2.56 | 2.14 |
| No education | 2.58 | 3.88 | 4.37 | 4.30 | 6.21 | 6.30 | 10.64 | 10.38 | 4.12 | 2.68 |
| **Household floor material** | | | | | | | | | | |
| Non-earth | 9.63 | 14.68 | 15.42 | 18.93 | 30.97 | 33.31 | 41.88 | 42.89 | 4.35 | 2.92 |
| Earth | 90.37 | 85.32 | 84.58 | 81.07 | 69.03 | 66.69 | 58.12 | 57.11 | 0.64 | 0.67 |
| **Toilet type** | | | | | | | | | | |
| VIP | 22.20 | 24.77 | 14.55 | 16.41 | 12.12 | 12.85 | 13.99 | 13.48 | 0.63 | 0.54 |
| Flush toilet | 9.19 | 11.43 | 9.16 | 11.40 | 14.80 | 18.84 | 26.90 | 31.00 | 2.93 | 2.71 |
| Pit | 42.16 | 37.59 | 62.77 | 59.74 | 56.57 | 53.32 | 49.68 | 48.67 | 1.18 | 1.29 |
| No-facility | 26.45 | 26.21 | 13.51 | 12.45 | 16.52 | 14.99 | 9.43 | 6.85 | 0.36 | 0.26 |
| **Source of water** | | | | | | | | | | |
| Tube-well | 92.87 | 92.49 | 91.56 | 88.73 | 80.47 | 79.14 | 83.80 | 84.64 | 0.90 | 0.92 |
| Non-tube-well | 7.13 | 7.51 | 8.44 | 11.27 | 19.53 | 20.86 | 16.20 | 15.36 | 2.27 | 2.05 |
| **Body mass index** | | | | | | | | | | |
| 18.5 to <25 | 47.70 | 46.32 | 57.11 | 57.13 | 59.85 | 60.20 | 56.78 | 52.39 | 1.19 | 1.13 |
| <18.5 | 49.70 | 49.88 | 34.30 | 31.75 | 24.15 | 20.47 | 11.39 | 10.30 | 0.23 | 0.21 |
| 25 to <30 | 2.27 | 2.48 | 7.20 | 9.33 | 13.43 | 15.08 | 25.74 | 28.20 | 11.34 | 11.37 |
| 30 and above | 0.34 | 1.31 | 1.39 | 1.79 | 2.57 | 4.25 | 6.09 | 9.11 | 17.91 | 6.95 |

1 Individual survey weight was used for each survey time points; VIP- ventilated improved pit

time in both the urban and rural areas, though several socio-demographic, economic and other health indicators have changed during the same observation period, including at least two times increase in total country's population [18].

Many potential factors had explored to understand the relationship with pregnancy termination. Maternal age and body mass index plays an important role, especially increasing age

**Table 2. Multivariable analysis[1] between different factors and fatal pregnancy termination at different time points among Bangladeshi women at the reproductive age group.**

| | 1997 | 2004 | 2011 | 2018 |
|---|---|---|---|---|
| **Age** | | | | |
| 19 to 29 years | Ref. | Ref. | Ref. | Ref. |
| <19 years | 0.32 (0.23–0.46) | 0.70 (0.57–0.86) | 0.74 (0.60–0.92) | 0.96 (0.78–1.17) |
| 30 to 39 years | 1.52 (1.32–1.76) | 1.36 (1.25–1.49) | 1.37 (1.26–1.48) | 1.40 (1.30–1.52) |
| 40 years and above | 1.63 (1.23–2.15) | 1.37 (1.24–1.52) | 1.39 (1.26–1.53) | 1.50 (1.37–1.65) |
| **Area** | | | | |
| Rural | Ref. | Ref. | Ref. | Ref. |
| Urban | 1.10 (0.86–1.41) | 1.07 (0.97–1.18) | 1.08 (0.99–1.17) | 1.03 (0.95–1.12) |
| **Wealth-index** | | | | |
| Poorest | Ref. | Ref. | Ref. | Ref. |
| Poorer | 0.96 (0.79–1.17) | 1.11 (0.98–1.25) | 1.01 (0.90–1.13) | 1.08 (0.98–1.20) |
| Middle | 1.06 (0.87–1.29) | 0.97 (0.85–1.11) | 1.08 (0.97–1.21) | 1.07 (0.95–1.20) |
| Richer | 0.93 (0.76–1.15) | 1.11 (0.97–1.27) | 1.10 (0.98–1.25) | 1.10 (0.95–1.27) |
| Richest | 0.92 (0.75–1.13) | 1.11 (0.92–1.32) | 1.30 (1.08–1.56) | 1.20 (1.00–1.44) |
| **Religion** | | | | |
| Muslim | Ref. | Ref. | Ref. | Ref. |
| Non- Muslim | 0.89 (0.70–1.14) | 0.88 (0.75–1.03) | 0.84 (0.74–0.94) | 0.90 (0.79–1.02) |
| **Education** | | | | |
| Higher | Ref. | Ref. | Ref. | Ref. |
| Up to secondary | 0.93 (0.79–1.10) | 1.01 (0.93–1.11) | 1.10 (1.01–1.21) | 1.11 (1.02–1.22) |
| Complete primary | 1.11 (0.90–1.35) | 1.04 (0.93–1.17) | 1.12 (1.01–1.24) | 1.08 (0.97–1.21) |
| No education | 1.16 (0.76–1.77) | 0.92 (0.78–1.09) | 0.95 (0.81–1.12) | 1.07 (0.93–1.22) |
| **Household floor material** | | | | |
| Non-earth | Ref. | Ref. | Ref. | Ref. |
| Earth | 0.71 (0.55–0.91) | 0.99 (0.85–1.16) | 1.18 (1.04–1.35) | 1.09 (0.97–1.22) |
| **Toilet type** | | | | |
| VIP | Ref. | Ref. | Ref. | Ref. |
| Flush toilet | 0.84 (0.63–1.10) | 0.94 (0.82–1.08) | 1.08 (0.96–1.22) | 1.09 (0.96–1.22) |
| Pit | 0.87 (0.73–1.04) | 0.92 (0.84–1.02) | 0.98 (0.89–1.08) | 1.04 (0.94–1.16) |
| No-facility | 0.98 (0.81–1.19) | 0.95 (0.81–1.11) | 1.08 (0.95–1.22) | 0.93 (0.78–1.11) |
| **Source of water** | | | | |
| Tube-well | Ref. | Ref. | Ref. | Ref. |
| Non-tube-well | 0.82 (0.66–1.02) | 1.14 (1.01–1.29) | 0.99 (0.89–1.09) | 1.01 (0.90–1.15) |
| **Body mass index** | | | | |
| 18.5 to <25 | Ref. | Ref. | Ref. | Ref. |
| <18.5 | 1.03 (0.91–1.17) | 0.97 (0.89–1.06) | 0.91 (0.84–1.00) | 1.02 (0.92–1.13) |
| 25 to <30 | 0.83 (0.53–1.30) | 1.04 (0.92–1.18) | 0.99 (0.90–1.09) | 1.06 (0.99–1.14) |
| 30 and above | 2.04 (1.37–3.04) | 0.99 (0.75–1.31) | 1.23 (1.05–1.44) | 1.27 (1.14–1.42) |

1 Individual survey weight was used for each survey time points to estimate the relative risk and 95% confidence interval;; VIP- ventilated improved pit

and obesity, increasing the risk of pregnancy termination across all survey time points. It is apparent that conception after 40 years considers a high-risk pregnancy, and thus the chance of miscarriage and stillbirth remains high [19, 20]. Several studies also determined that the risk of miscarriage increased after 30 years, and at least half of the conceptions were miscarriage after 45 years [19, 20]. Overweight and obese pregnant women are at greater risk of preterm

birth, stillbirth and neonatal death than women with normal body weight before pregnancy and have optimal pregnancy weight gain [9, 10]. Placental dysfunction is the primary triggering factor resulting from a change in the inflammatory process due to obesity [9, 10]. Additionally, maternal blood glucose level might play an integral role, particularly in the risk of gestational diabetes and the development of insulin resistance in utero [21]. In Bangladesh, the proportion of women with overweight and obese increased from 12% in 2007 to 32% in 2018 [22]. It is now high time to strengthen the future preventive strategies for overweight and obese in a highly populated country like Bangladesh.

The current study did not find any association with the earth as household floor materials, toilet facilities and drinking water sources. Nevertheless, all are related to infection, especially soil-born disease and other infectious diseases transmitted to the faecal-oral route [23, 24]. However, the study was not designed to distinguish between different types of a fatal termination of pregnancy, especially stillbirth and spontaneous abortion or miscarriage, where the aetiologies related to infection were in many instances.

A relatively weak association was observed between maternal education to the secondary level. Women's education has been essential for making timely decisions to prevent adverse maternal and foetal outcomes [7]. Such a relationship might explain by other factors related to maternal education, such as employment. Women empowerment plays an integral part in reducing fatal pregnancy termination because it maximises their pre-pregnancy health, access family planning for timing of their pregnancies and demand and engages in high-quality antenatal and intrapartum care [2]. The current study did not assess women employment. However, in the last two decades, the overall county's women's education and engagement with income-generating sources have increased [16]. Thus, the fatal pregnancy termination relationship might be related to a lack of proper rest, especially in the first trimester. Such an explanation might apply to an association between the most affluent wealth index and fatal pregnancy termination. However, we did not differentiate either woman from the wealthiest wealth index engaged in income-generating activities and contributing to overall family wealth.

## Policy and future implications

The unchanged proportion of fatal pregnancy termination would curtail the antenatal policy for pregnant women in Bangladesh. Over the past few decades, many government and non-government investments have been made to improve maternal health during pregnancy through social and cultural motivation and active participation and engagement, reduce the possible barriers, ensue optimal service and safe delivery in the health facilities [25–28]. Despite all these efforts and progress, essential care is not yet optimised for all pregnancies. Research scarcity in the prevention of fatal termination of pregnancy remains a significant concern not only just due to lack of resources but also lack of solid research collaboration between public and private sectors, no centralised monitoring and surveillance system, existing cultural and social stigmata and traditional caring attitude and practice are potential barriers. Mothers, irrespective of their residents, consistently had almost similar risks in all survey points. It does not indicate that there is no difference between urban and rural areas. However, the wind of development is extending more in rural areas because of the extension of big cities, and the patterns of morbidities are notably different. Thus, comprehensive nationwide collaborative efforts are essential for further detailed investigation, sustained monitoring and evaluation for effective functioning of the existing ante-natal health system and policies, identifying and save at-risk pregnancy through appropriate interventions, monitoring risk factors, and ensuring safe-deliver and post-partum health to prevent fatal pregnancy termination.

## Strengths and limitations

The study had several strengths and limitations. The main strengths were a large nationally representative sample and two decades of observation (1997 to 2018) following identical study design and study tools. The other strength was that estimating the relative risk over odds ratio would be more meaningful to determine the risk of a fatal pregnancy termination with different explanatory factors over time. However, the study's primary limitation was not distinguishing how the fatal pregnancy termination happened either by miscarriage or abortion or stillbirth or by menstrual regulation. Thus, the survey design did not aim to determine the risk factors for pregnancy termination which need to explore in detail with future research endower. Unable to assess the relationship with livestock recorded at the household level might be another limitation. Moreover, several other genetic and hormonal [5, 29] and climate factors such as arsenic exposure and inter-outcome intervals after a stillbirth/abortion/miscarriage [30] might be associated with the pregnancy termination, which remains beyond the capacity to assess from the current BDHS surveys.

## Conclusion

From 1997 to 2018 in Bangladesh, the overall proportion of ever pregnancy termination remains almost static with minor variations. However, the risk factors of pregnancy termination changed over time. Therefore, compared to the global trend with the increasing country's population, it is essential to understand further the different types and reasons for fatal pregnancy termination for better women's reproductive health and successful pregnancy outcome or live birth.

## Acknowledgments

The authors thank MEASURE DHS for the permission to use the Bangladesh Demographic and Health Survey data from 1997 to 2018. We also acknowledge the collaboration of the National Institute of Population Research and Training (NIPORT), ICF International (USA), and Mitra and Associates in conducting the Bangladesh Demographic and Health Surveys.

## Author Contributions

**Conceptualization:** Shathi Das, Sumon Kumar Das.

**Data curation:** Jui Das, Sumon Kumar Das.

**Formal analysis:** Shathi Das, Jui Das, Palash Roy, Sumon Kumar Das.

**Methodology:** Shathi Das, Jui Das, Kamanasis Mazumder, Rehana Begum, Sumon Kumar Das.

**Supervision:** Sumon Kumar Das.

**Writing – original draft:** Shathi Das.

**Writing – review & editing:** Jui Das, Kamanasis Mazumder, Palash Roy, Rehana Begum, Sumon Kumar Das.

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
