## [Decision Letter · Decision Letter 0]

12 Oct 2021

PONE-D-21-25908Trend and risk factors of fatal pregnancy termination: A long-term nationwide population-based cross-section survey in BangladeshPLOS ONE

Dear Dr. Das,

Thank you for submitting your manuscript to PLOS ONE. After careful consideration, we feel that it has merit but does not fully meet PLOS ONE’s publication criteria as it currently stands. Therefore, we invite you to submit a revised version of the manuscript that addresses the points raised during the review process.

We look forward to receiving your revised manuscript.

Kind regards,

Ashraf Dewan, PhD

Academic Editor

PLOS ONE

Journal Requirements:

"The funders had no role in study design, data collection and analysis, decision to publish, or preparation of the manuscript"

3. We note that you have stated that you will provide repository information for your data at acceptance. Should your manuscript be accepted for publication, we will hold it until you provide the relevant accession numbers or DOIs necessary to access your data. If you wish to make changes to your Data Availability statement, please describe these changes in your cover letter and we will update your Data Availability statement to reflect the information you provide

Additional Editor Comments (if provided):

I have now received comments on your submission. Based on two reviewers comments, I now invite you revise you work.

Reviewers' comments:

Reviewer's Responses to Questions

**Comments to the Author**

1. Is the manuscript technically sound, and do the data support the conclusions?

Reviewer #1: Yes

Reviewer #2: Yes

2. Has the statistical analysis been performed appropriately and rigorously? 

Reviewer #1: Yes

Reviewer #2: I Don't Know

3. Have the authors made all data underlying the findings in their manuscript fully available?

Reviewer #1: Yes

Reviewer #2: Yes

4. Is the manuscript presented in an intelligible fashion and written in standard English?

Reviewer #1: Yes

Reviewer #2: Yes

5. Review Comments to the Author

Reviewer #1: Dear Authors;

It would be appropriate to write the expressions in the results section in a simpler and more appropriate language. I think that this manuscript can be published in your journal.

Good works.

Reviewer #2: Overall the manuscript is interesting. However, few of the correction in spelling and further review of the result section and discussion are necessary.

Abstract: The introduction part of the abstract should rewrite.

Introduction: you have written reference in the bracket. Please provide reference number.

Method: In the methodology section, it seems you have used some results. Please review it.

Result: Result section is not clear in few points; for instance the multivariable analysis. Researchers should review further.

Discussion: The discussion section has been written detail, but it should mention here about the policy implication and further implication of this analysis.

6. PLOS authors have the option to publish the peer review history of their article (what does this mean?). If published, this will include your full peer review and any attached files.

Reviewer #1: No

Reviewer #2: No

---

## [Author Response · Author response to Decision Letter 0]

1 Dec 2021

Date: 01 December 2021

To

Ashraf Dewan 

Academic Editor

PLOS ONE

Subject: Resubmission of manuscript: Trend and risk factors of fatal pregnancy termination: A long-term nationwide population-based cross-section survey in Bangladesh (PONE-D-21-25908) 

Dear Sir,

Many thanks for sharing reviewers’ comments with regard to the manuscript “Trend and risk factors of fatal pregnancy termination: A long-term nationwide population-based cross-section survey in Bangladesh (PONE-D-21-25908)“. Indicated below are the point by point responses to the comments and suggestions made by the reviewers for your consideration. All the necessary changes have been incorporated in the manuscript, tables and figure in the track change version and both clean and track change version have been uploaded accordingly. We hope that our response will be appropriate to qualify the manuscript for publication in your well reputed journal. 

Comments to the Author

 Is the manuscript technically sound, and do the data support the conclusions?

Reviewer #1: Yes

Remarks: No remarks. Thank you for your positive feedback.

Reviewer #2: Yes

Remarks: No remarks. Thank you for your positive feedback.

2. Has the statistical analysis been performed appropriately and rigorously?

Reviewer #1: Yes

Remarks: No remarks. Thank you for your positive feedback.

Reviewer #2: I Don't Know

3. Have the authors made all data underlying the findings in their manuscript fully available?

Reviewer #1: Yes

Remarks: No remarks. Thank you for your positive feedback.

Reviewer #2: Yes

Remarks: No remarks. Thank you for your positive feedback.

4. Is the manuscript presented in an intelligible fashion and written in standard English?

Reviewer #1: Yes

Remarks: No remarks. Thank you for your positive feedback.

Reviewer #2: Yes

Remarks: No remarks. Thank you for your positive feedback.

5. Review Comments to the Author

Reviewer #1: Dear Authors;

It would be appropriate to write the expressions in the results section in a simpler and more appropriate language. I think that this manuscript can be published in your journal.

Good works.

Remarks: Thank you for your positive feedbacks and concerns. Necessary modifications have been done as suggested in plain language. 

Reviewer #2: Overall the manuscript is interesting. However, few of the correction in spelling and further review of the result section and discussion are necessary.

Abstract: The introduction part of the abstract should rewrite.

Remarks: Thank you. The Introduction of the Abstract has been rewritten. 

Introduction: you have written reference in the bracket. Please provide reference number.

Remarks: Thank you. The reference has been cited now. 

Method: In the methodology section, it seems you have used some results. Please review it.

Remarks: Thank you. All result related information has been moved to the result section as appropriate. 

Result: Result section is not clear in few points; for instance the multivariable analysis. Researchers should review further.

Remarks: Thank you. Necessary modifications have been done accordingly, especially the multivariate analysis. 

Discussion: The discussion section has been written detail, but it should mention here about the policy implication and further implication of this analysis.

Remarks: Thank you so much for your practical concerns. A separate section has been incorporated under policy implication under the discussion section with a sub-heading of “Police implication and future implication" with relevant citations (ref. 25-28)

Moreover, some minor editing also done where required. 

Sincerely yours, 

Shathi Das

St. Gregory's High School & College, 

82, Municipal Office Street 

Luxmibazar, Dhaka-1100, Bangladesh

Phone: +88 01873297467

Email: dasshathi1@gmail.com

Sumon Kumar Das 

Centre for Child Development and Education, 

Menzies School of Health Research, 

Charles Darwin University

Building Red 9, Casuarina Campus

PO Box 41096, Casuarina NT 0811, Australia

Email: sumon.das@menzies.edu.au; dassumonkumar1@gmail.com

---

## [Decision Letter · Decision Letter 1]

20 Jan 2022

Trend and risk factors of fatal pregnancy termination: A long-term nationwide population-based cross-section survey in Bangladesh

PONE-D-21-25908R1

Dear Dr. Das,

We’re pleased to inform you that your manuscript has been judged scientifically suitable for publication and will be formally accepted for publication once it meets all outstanding technical requirements.

Kind regards,

Ashraf Dewan, PhD

Academic Editor

PLOS ONE

Additional Editor Comments (optional):

Reviewers' comments:

Reviewer's Responses to Questions

**Comments to the Author**

1. If the authors have adequately addressed your comments raised in a previous round of review and you feel that this manuscript is now acceptable for publication, you may indicate that here to bypass the “Comments to the Author” section, enter your conflict of interest statement in the “Confidential to Editor” section, and submit your "Accept" recommendation.

Reviewer #1: All comments have been addressed

Reviewer #2: All comments have been addressed

2. Is the manuscript technically sound, and do the data support the conclusions?

Reviewer #1: Yes

Reviewer #2: Yes

3. Has the statistical analysis been performed appropriately and rigorously? 

Reviewer #1: Yes

Reviewer #2: Yes

4. Have the authors made all data underlying the findings in their manuscript fully available?

Reviewer #1: Yes

Reviewer #2: Yes

5. Is the manuscript presented in an intelligible fashion and written in standard English?

Reviewer #1: Yes

Reviewer #2: Yes

6. Review Comments to the Author

Reviewer #1: Dear Authors;

I re-evaluated the manuscript named "Trend and risk factors of fatal pregnancy termination: A

long-term nationwide population-based cross-section survey in Bangladesh", which I had previously evaluated and made recommendations. I think that this manuscript can be published in this journal.

Good works.

Reviewer #2: Overall, the manuscript is good and has been organized well. It has been improved a lot after the first revision. I would suggest to check the spelling mistakes throughout the manuscript.

In the result section, you have mentioned the association of education and risk factors. If possible, put some indicators.

7. PLOS authors have the option to publish the peer review history of their article (what does this mean?). If published, this will include your full peer review and any attached files.

Reviewer #1: No

Reviewer #2: No
